# The life expectancy of older couples and surviving spouses

**Janice Compton**[1☯], **Robert A. Pollak**[2,3,4☯] *

**1** Department of Economics, University of Manitoba, Winnipeg, Manitoba, Canada, **2** Department of Economics and Olin School of Business, Washington University in St. Louis, St. Louis, Missouri, United States of America, **3** National Bureau of Economic Research, Cambridge, Massachusetts, United States of America, **4** IZA Institute of Labor Economics, Bonn, Germany

☯ These authors contributed equally to this work.
* pollak@wustl.edu

**Data Availability Statement:** The data underlying this study are publicly available from the following sources: Census data: Ruggles S, Genadek K, Goeken R, Grover J, Sobek M. Integrated Public Use Microdata Series: Version 7.0 [dataset]. Minneapolis: University of Minnesota, 2017.

## Abstract

Individual life expectancies provide information for individuals making retirement decisions and for policy makers. For couples, analogous measures are the expected years both spouses will be alive (joint life expectancy) and the expected years the surviving spouse will be a widow or widower (survivor life expectancy). Using individual life expectancies to calculate summary measures for couples is intuitively appealing but yield misleading results, overstating joint life expectancy and dramatically understating survivor life expectancies. This implies that standard "individual life cycle models" are misleading for couples and that "couple life cycle models" must be substantially more complex. Using the CDC life tables for 2010, we construct joint and survivor life expectancy measures for randomly formed couples. The couples we form are defined by age, race and ethnicity, and education. Due to assortative marriage, inequalities in individual life expectancies are compounded into inequalities in joint and survivor life expectancies. We also calculate life expectancy measures for randomly formed couples for the 1930–2010 decennial years. Trends over time show how the relative rate of decrease in the mortality rates of men and women affect joint and survivor life expectancies. Because our couple life expectancy measures are based on randomly formed couples, they do not capture the effects of differences in spouses' premarital characteristics (apart from sex, age, race and ethnicity, and, in some cases, education) or of correlations in spouses' experiences or behaviors during marriage. However, they provide benchmarks which have been sorely lacking in the public discourse.

## Introduction

Measures of joint and survivor life expectancy are potentially useful to those designing or evaluating policies affecting older couples and to couples making retirement, savings, and long-term care decisions. However, couple-based measure of life expectancies are virtually unknown in the social sciences. In discussions of intra-household decision making (e.g., regarding the timing of retirement and the claiming of retirement and social security benefits),

https://doi.org/10.18128/D010.V7.0. Mortality rates: CDC, National Center for Health Statistics. Life Tables. https://www.cdc.gov/nchs/products/life_tables.htm. August 3, 2017. Mortality rates by education: Bound J, Geronimus AT, Rodriguez JM, Waidmann TA. Measuring recent apparent declines in longevity: the role of increasing educational attainment. Health Affairs. 2015;34(12):2167-73.

**Funding:** RAP G-B2012-24 Alfred P. Sloan Foundation https://sloan.org/ The funders had no role in study design, data collection and analysis, decision to publish, or preparation of the manuscript.

**Competing interests:** The authors have declared that no competing interests exist.

the difference between the life expectancies of a wife and husband is sometimes treated as an informal measure of survivor life expectancy [1–3]. As we shall show, neither couple life expectancy nor survivor life expectancies can be inferred from individual life expectancies. This implies that the standard "individual life cycle models" widely used in economics are misleading for couples and that "couple life cycle models" are substantially more complex.

We illustrate the construction of measures of joint and survivor life expectancy for a randomly formed non-Hispanic white couple in which the wife was 60 and the husband 62 in 2010—that is, the wife was born in 1950 and her husband in 1948. We focus on 60-year-old wives and their husbands because these are ages at which many couples make crucial retirement-related decisions such as leaving career employment and claiming retirement and social security benefits. Thus, these are ages at which we would expect joint and survivor life expectancies to be especially salient.

Census data show that in 2010 the average age gap between 60-year-old non-Hispanic white women and their husbands was about 2 years. The 2010 CDC life tables show that the life expectancy of a 60-year-old non-Hispanic white woman was 24.4 years and that of a 62-year-old non-Hispanic white man was 20.2 years. The intuitions that the spouses will die at about the same time (e.g., within 4 or 5 years of each other) and that the wife will not live for a long time after her husband's death are incorrect. Only if the mortality distributions of the husband and the wife were so tightly concentrated that they did not overlap (e.g., if 60 year old wives always lived longer than their 62 year old husbands) would it would be correct to use the minimum of husband and wife life expectancies as the joint life expectancy and to use the difference between the individual life expectancies as the survivor life expectancy. And, as intuition suggests, if the overlap between the mortality distributions were small, then these measures would be good approximations. But because the overlap is substantial, calculations based on individual life expectancies provide poor approximations of joint life expectancy and very poor approximations of survivor life expectancy.

Although Goldman and Lord [4] proposed couple life expectancy measures more than three decades ago, these measures have not been widely discussed or adopted outside the actuarial literature; see, for example, Bowers et al. [5] Brown and Poterba [6] investigate an alternative notion of "joint life expectancy"–the number of years that at least one spouse is expected to be alive. Similarly, the Internal Revenue Service (IRS) provides race-ethnicity-gender-neutral tables to enable taxpayers to calculate the tax liabilities associated with joint life annuities and distributions from Individual Retirement Accounts (IRAs) [7]. Demographers, economists, gerontologists, and sociologists, however, have generally focused on age-specific mortality rates rather than the summary measures of joint and survivor life expectancies [8, 9]. Although age-specific mortality rates are the basic building blocks of life expectancy measures, they are much less accessible than summary measures such as the probability that the wife will predecease the husband and, we argue, much less accessible than joint and survivor life expectancy.

Using male and female mortality rates from the CDC National Center for Health Statistics (NCHS), we calculate the joint life expectancy of randomly formed older couples and the life expectancy of surviving spouses. These calculations illustrate the extent to which approximations of couple life expectancies based on individual life expectancies are seriously misleading.

We do this separately for non-Hispanic whites, blacks, and Hispanics and, within racial groups for 2010, by educational attainment. For example, the joint life expectancies of couples consisting of 60 year old women and their husbands range from a low of 13.61 years for a black couple in which neither spouse has a college degree to a high of 18.99 years for a non-Hispanic white couple in which both spouses have college degrees. The measures that condition on educational attainment show that the differences in couple life expectancy by education is greater than the racial differences. We also calculate joint and survivor life expectancy

for each census year beginning in 1930, and analyze trends in joint and survivor life expectancy from 1930 to 2010. The measures over the decennial years show how the relative declines in the mortality rates of men and women led to increases in both joint life expectancy and the number of years women could expect to be widows.

Our use of randomly formed couples provides measures that are analogous to individual life expectancies. They are summary measures that do not reflect individual differences, but rather provide informative benchmarks. Our decision to base the calculations on easily accessible data requires certain simplifications: (1) we calculate joint and survivor life expectancies from individual age-specific mortality rates; and (2) we use the individual mortality rates calculated from the full population, not from the married populations. These simplifications imply that we cannot take account of the correlation in mortality rates between husbands and wives, except to the extent that they are captured by educational attainment. We discuss each of these in turn.

Our three simplifications ensure that the measures presented here are analogous to standard individual life expectancy measures. First, like the measures presented here, individual life expectancy measures are typically based on cross-section (i.e., period) mortality rates. This assumes stability in age-specific mortality rates, an assumption that is well understood to be improbable. With longitudinal data, researchers can estimate the rate of change in age specific mortality rates and incorporate these forecasts into the life expectancies.

Second, evidence suggests that the age-specific mortality rates of married individuals are less than those of unmarried individuals both because healthier individuals select into marriage and because marriage has protective effects [10–14]. Our calculations, which are based on NCHS life tables and randomly formed couples, ignore these effects and assume that individuals' age-specific mortality rates are independent of their marital status. It would be erroneous to calculate joint and survivor life expectancies based on the age-specific mortality rates of married individuals because doing so assumes that married individuals remain married until death. Clearly, this is rarely true for both spouses and so these calculations will overestimate joint life expectancy. Moreover, it is not clear the extent to which mortality rates based on marital status reflect other attributes–education, race, place of birth, income–that may be related to both mortality and marital status. Therefore, to account for marital status and the protective effects of marriage when calculating joint and survivor life expectancies requires detailed longitudinal data on couples. We leave this to further research.

These two simplifications imply that we are ignoring the correlation in mortality rates of husbands and wives. Although we expect assortative marriage, shared environments, and behavioral habits to create correlations in these mortality rates, we base our calculations on the age-specific mortality of individuals, randomly assigned to form couples based on age and race. The effect of this is to impose the assumption that spouses' mortality rates are independent of each other. Just as an individual's life expectancy will differ from the population estimate due to their characteristics, experiences, and life choices, a couple's joint and survivor life expectancies will differ from the population estimate due to their characteristics, experiences, and choices.

Overall, the calculations of joint and survivor life expectancies based on age-specific mortality rates of men and women–for example, 60 year old non-Hispanic white women randomly matched with 62 year old non-Hispanic white men—certainly do not capture all the nuances of mortality and marriage. They do, however, provide benchmarks which have been sorely lacking in the public discourse. For example, Holden and Kuo [15] analyze responses to questions on expected mortality in the Health and Retirement Study (HRS) and find that, on average, both spouses overestimate their joint years remaining and underestimate their likelihood

of being widowed. A better understanding of the joint and survivor life expectancies may aid couples in decision-making.

## Method

The calculation of joint and survivor life expectancies requires the age-specific mortality distributions, not individual life expectancies. We use the NCHS life tables [16] for men and women to construct mortality distributions and life tables for randomly constructed couples and from these we calculate joint life expectancy (i.e., the expected number of years that both spouses will live). We then calculate the probability that the wife (husband) will die at each age and use the individual life tables to calculate survivor life expectancy conditional on the sex of the surviving spouse (e.g., if the wife is the surviving spouse, we calculate her expected number of years as a widow). Although we use the terms "widow" and "widower," we actually calculate the expected number of years that both spouses are alive, regardless of whether or not they remain married, and how long a surviving spouse can expect to live after the death of his or her current spouse, regardless of remarriage.

The construction of mortality distributions for couples is straightforward but tedious. To illustrate, we continue to focus on the case in which we form a couple in which the wife was 60 and the husband 62 in 2010. From the individual life tables for men and for women, we calculate the probability that one or both spouses will die in 2010. This probability is the sum of the probabilities of three mutually exclusive events:

1. the husband will die between 62 and 63 AND the wife will not die between 60 and 61

2. the wife will die between 60 and 61 AND the husband will not die between 62 and 63, and

3. the husband will die between 62 and 63 AND the wife will die between 60 and 61.

The sum of these three probabilities is, of course, equal to one minus the probability that neither spouse will die in 2010. Thus, if our only aim were to calculate joint life expectancy, it would be easier to focus on the probability that neither spouse would die at each age or in each year. The drawback of proceeding in this way is that to calculate survivor life expectancies we must first calculate the probability that the wife (husband) will become a widow (widower) at each age.

For couples that survive into 2011, we proceed in the same way, calculating the probability that the husband will die between 63 and 64 and the wife will not die between 61 and 62, etc. These calculations give "mortality rates" for the couple for each year, and from these we can construct a "couple life table." More specifically, beginning with a cohort of 100,000 couples with the wife aged 60 and the husband aged 62, we can calculate expected transitions to widows, widowers, and "couple death" in each year. This corresponds to the $\ell_X$ and $d_X$ columns in standard individual life tables, with the subscript denoting year forward rather than age.

From the couple life table, we calculate the couple's joint life expectancy by applying to couples the standard life expectancy calculation typically applied to individuals. For our focal couple–a non-Hispanic white couple in which the wife was 60 and the husband 62 in 2010 –joint life expectancy is 17.7 years (recall that her individual life expectancy is 24.4 years and his individual life expectancy is 20.2 years).

Survivor life expectancy answers questions like: "If the wife is the surviving spouse, how many years can she expect to live after her husband's death?" The survivor life expectancies are appropriately weighted averages of individual life expectancies at each age, where the weights are the probabilities of couple death in each year, conditional on couple survival to that year and conditional on the sex of the surviving spouse.

The probability that our focal 60-year-old wife will predecease her 62-year-old husband is 0.37, a surprisingly high probability that reflects the substantial overlap of their mortality distributions. Hence, the probability that the wife will be the surviving spouse is 0.63 and, if she is the surviving spouse, her survivor life expectancy is 12.5 years. If the husband is the surviving spouse, his survivor life expectancy is 9.5 years.

This example demonstrated the calculation for a non-Hispanic white 60-year-old woman married to a non-Hispanic white 62-year-old man. To generalize the calculations to 60-year-old women married to men of each age, we use the 2010 Census [17] to determine the proportion of 60-year-old women married to men of each age, and use these proportions as weights. Thus the 60 year old woman is the anchor of each calculation: her joint and survivor life expectancies are a weighted average of the joint and survivor life expectancies she would face if married to a man of each age, and the weights are the probability that she is married to a man of each age. Thus, the differences we report in joint and survivor life expectancy across groups reflect differences in both age-specific mortalities and in the age gaps between 60-year-old women and their husbands.

We repeat this calculation for black and Hispanic women in 2010 and for black and white women by education of self and spouse in 2010. To calculate the joint and survivor life expectancies by couple education, we use data from Bound et al. [18] since the NCHS life tables do not condition on education. Finally, we use the decennial census data from 1930–2010 to calculate the joint and survivor life expectancies for 60-year-old white and black women over time.

## Results

### Joint and survivor life expectancies by race and ethnicity

Table 1 shows the 2010 measures of life expectancy for couples formed at random based on the population age profiles of non-Hispanic white, black, and Hispanic couples. We combine Hispanic blacks with blacks rather than with Hispanics. We define the race and ethnicity of couples by the race and ethnicity of the wife, that is, we consider only same-race couples in the

**Table 1. Life expectancy measures, 2010.** By race and ethnicity, wife is aged 60.

| | Non-Hispanic White | Black | Hispanic |
|---|---|---|---|
| **Wife Life Expectancy** | 24.40 | 23.05 | 26.40 |
| | (0.00) | (0.00) | (0.00) |
| **Husband Life Expectancy** | 20.17 | 18.22 | 21.05 |
| | (3.59) | (3.74) | (4.59) |
| **Joint Life Expectancy** | 17.66 | 15.45 | 18.79 |
| | (2.08) | (2.12) | (2.82) |
| **Survivor Life Expectancy (Wife)** | 12.48 | 13.52 | 13.13 |
| | (1.21) | (1.19) | (1.77) |
| **Survivor Life Expectancy (Husband)** | 9.48 | 10.05 | 9.34 |
| | (1.51) | (1.77) | (1.85) |
| **Probability that Wife is the Surviving Spouse** | 0.63 | 0.63 | 0.65 |
| | (0.10) | (0.10) | (0.12) |
| **Age Gap (Husband–Wife)** | 1.91 | 1.81 | 2.25 |
| | (4.79) | (5.61) | (5.90) |
| **Sample Size** | 10,967 | 848 | 620 |

Calculations by authors. Standard errors in parentheses.

calculation. There is little intermarriage reported in the 2010 census for non-Hispanic white women or for black women aged 60. Fully 96 percent of married non-Hispanic white women are married to non-Hispanic white men, and over 91 percent of married black women are married to black men. Hispanic women are more likely to intermarry; in 2010, 76 percent of married Hispanic women were married to Hispanic men. We present the individual life expectancy of women aged 60, the average life expectancy of spouses, and measures of joint and survivor life expectancy for the appropriately weighted average of couples in which the wife was aged 60.

Both Hispanic men and Hispanic women have longer life expectancies than their non-Hispanic white and black counterparts, and Hispanic couples have a higher joint life expectancy. We have taken at face value the mortality distributions and life expectancies for Hispanic men and women reported by the NCHS, but these should be viewed with caution. The NCHS life tables report that Hispanics have substantially longer life expectancies than non-Hispanic whites, despite Hispanics' lower incomes and lower levels of educational attainment. Demographers have termed this the "Hispanic Paradox" and have offered a variety of explanations [19, 20]. Black et al. [21] argue that the longer Hispanic life expectancies are probably due to measurement error.

Black men and black women have shorter life expectancies than their non-Hispanic white and Hispanic counterparts, and black couples have a lower joint life expectancy. The pattern for survivor life expectancy is the opposite: conditional on becoming a widow, black women have a longer survivor life expectancy (13.5 years) than Hispanic women (13.1 years) and non-Hispanic white women (12.5 years). This pattern reflects the larger gap in the mortality rates of black men and women compared to Hispanic and white men and women. Hispanic men have the shortest survivor life expectancies (9.3 years), followed by white men (9.5 years), with black men having the longest expected survivor life expectancies (10.1 years). While Hispanic men and women have the longest individual life expectancies at age 60, they have the lowest expected number of years in widowhood. black men and women have the shortest life expectancies at age 60, and they have the highest expected number of years in widowhood.

## Joint and survivor life expectancies by education

We now discuss the association between these racial and ethnic differences in joint and survivor life expectancy and education, using mortality estimates from Bound et al. [18] to calculate life expectancy measures by education for non-Hispanic white couples and black couples. Bound et al. [18] calculate mortality rates for four education categories—less than high school, high school graduate, some college, and college graduate—for non-Hispanic whites and for blacks, conditional on survival to age 25. We have extended the estimates in two ways. First, we convert their five-year estimates to one-year age estimates using a cubic spline. Second, we extend their estimates for ages above age 85 using the proportional increases in mortality for ages above 86 in the full NCHS data.

We classify randomly formed couples into four education categories, using a "power couples" terminology [22, 23]. Couples are defined as "low-power" if neither the man or woman has a college degree; "half-power-her" if only the woman has a college degree; "half-power-him" if only the man has a college degree; and "full-power" if both have college degrees. The estimates from Bound et al. [18] are based on administrative data that have been criticized for underestimating education, especially at the lower end of the education distribution (Rostron [24], Hendi [25], and Sasson [26]). Because we focus on the differences between college graduates and non-college graduates, concerns about the administrative data used in Bound are less relevant.

**Table 2. Life expectancy measures, 2010.** By education, wife is aged 60.

| | Non-Hispanic White Couples | | | | | Black Couples | | | | |
|---|---|---|---|---|---|---|---|---|---|---|
| | Low Power | Half Power (Her) | Half Power (Him) | Full Power | All | Low Power | Half Power (Her) | Half Power (Him) | Full Power | All |
| **Wife Life Expectancy** | | | | | | | | | | |
| | 23.68 | 26.07 | 24.73 | 26.07 | 24.61 | 21.70 | 23.79 | 22.51 | 23.79 | 22.26 |
| | (1.79) | (0.00) | (1.84) | (0.00) | (1.83) | (1.43) | (0.00) | (1.50) | (0.00) | (1.53) |
| **Husband Life Expectancy** | | | | | | | | | | |
| | 18.71 | 19.82 | 22.06 | 22.12 | 20.13 | 16.95 | 17.46 | 19.45 | 19.76 | 17.57 |
| | (4.17) | (4.39) | (3.47) | (3.46) | (4.25) | (4.36) | (4.22) | (3.77) | (3.68) | (4.36) |
| **Joint Life Expectancy** | | | | | | | | | | |
| | 15.53 | 17.17 | 18.31 | 18.99 | 16.91 | 13.61 | 14.84 | 15.59 | 16.39 | 14.26 |
| | (3.04) | (3.42) | (2.52) | (2.82) | (3.32) | (2.68) | (2.59) | (2.23) | (2.26) | (2.78) |
| **Survivor Life Expectancy (Wife)** | | | | | | | | | | |
| | 13.24 | 13.50 | 11.70 | 11.93 | 12.71 | 13.40 | 13.99 | 12.61 | 12.87 | 13.33 |
| | (1.58) | (1.79) | (1.36) | (1.38) | (1.68) | (1.45) | (1.52) | (1.36) | (1.44) | (1.49) |
| **Survivor Life Expectancy (Husband)** | | | | | | | | | | |
| | 9.69 | 8.96 | 10.07 | 9.27 | 9.59 | 9.89 | 9.36 | 10.47 | 10.05 | 9.90 |
| | (1.71) | (1.48) | (1.43) | (1.28) | (1.59) | (2.06) | (1.70) | (1.76) | (1.42) | (1.94) |
| **Probability that Wife is the Surviving Spouse** | | | | | | | | | | |
| | 0.65 | 0.68 | 0.59 | 0.63 | 0.64 | 0.64 | 0.68 | 0.59 | 0.62 | 0.64 |
| | (0.12) | (0.12) | (0.11) | (0.10) | (0.12) | (0.12) | (0.12) | (0.11) | (0.10) | (0.12) |
| **Age Gap (Husband–Wife)** | | | | | | | | | | |
| | 1.96 | 1.93 | 1.89 | 1.83 | 1.91 | 1.47 | 2.28 | 2.79 | 2.59 | 1.81 |
| | (5.14) | (5.16) | (4.18) | (4.20) | (4.79) | (5.81) | (5.10) | (5.31) | (4.85) | (5.61) |
| **Sample** | 5,828 | 994 | 1,703 | 2,442 | 10,967 | 573 | 89 | 72 | 114 | 848 |

Calculations by authors. Standard errors in parentheses.

Table 2 presents the life expectancy measures by education, with couples formed using weighted averages calculated from 2010 Census data. For both men and women, education is associated with lower mortality and substantially greater life expectancy [18, 27–29]. We do not speculate on what portion of this association is causal and what portion reflects correlations between education and other factors, both genetic and environmental. Due to positive assortative marriage on education, the education differences in individuals' life expectancies translate into substantial differences in joint and survivor life expectancies. Consider first the joint life expectancy measure. For both non-Hispanic white couples and black couples, as we move from low-power couples, to half-power-her, to half-power-him, to full power couples, joint life expectancy increases steadily. For non-Hispanic white couples, joint life expectancy increases across education categories from 15.5 to 19.0 years; for black couples, joint life expectancy increases from 13.6 to 16.4 years.

While joint life expectancies are approximately three years longer for non-Hispanic white couples in each category, survivor life expectancies are slightly higher for black men and women compared with non-Hispanic white men and women. For non-Hispanic white couples and for black couples, the husband survivor life expectancy is highest when he has a college degree and she does not (10.1 and 10.5 years). For these half-power-him couples, his life expectancy is higher than average and hers is lower than average, which results in a higher expected length of widowerhood for the husband. The husband survivor life expectancy is lowest when she has a college degree and he does not (9.0 years for non-Hispanic white couples and 9.4 years for black couples). For these half-power-her couples, his life expectancy is lower than

average and hers is higher than average, resulting in lower expected years of widowerhood for the husband.

We find that the differences in life expectancies by education are, in all cases, greater than the racial differences. Similarly, the difference in the probability that the husband dies first is greater across education categories than the differences between blacks and non-Hispanic whites. For black couples and for non-Hispanic white couples, the probability that the wife will die first is lowest for couples in which only the husband has a college degree (half-power-him couples, at 0.59), and highest for couples in which only the wife has a college degree (half-power-her couples, at 0.68). Although these numbers reflect both differences in life expectancy and age gap across education groups, this pattern continues to hold when we fix the age gap at two years and consider only differences in life expectancies between blacks and non-Hispanic whites.

## Joint and survivor life expectancies over time

We next describe the trends in joint and survivor life expectancies for white and black couples randomly formed using 1930–2010 Census data. We begin in 1930 because the mortality data for older individuals from 1920 and before is not comparable with the data from 1930 and after. The difficulty is that prior to 1930, the survivor tables end at age 89 (i.e., the probability of death between ages 89 and 90 is one). The tables from 1930 onwards extend to 101. The NCHS life tables for earlier census years do not include mortality distributions for Hispanics. As previously, we analyze couples consisting of 60 year old women and a weighted average of men at each age, with the weights reflecting the population of couples in the data. Thus, the changes we report in joint and survivor life expectancy between 1930 and 2010 reflect both changes in age-specific mortalities and changes in the age gaps between spouses. The gender differences in age-specific mortality have varied throughout the century. For a discussion of the determinants of the gap, see Goldin and Lleras-Muney [30].

Individual life expectancy for 60 year old women has increased steadily over the decades, and the gap between white and black women has remained fairly constant. The individual life expectancy of men married to 60 year old women increased both because the age gap has fallen (so husbands in 2010 are on average younger than husbands in 1930) and because the life expectancy of older men has increased. In Fig 1 we present joint and survivor life expectancies for randomly formed white couples in which the wife was aged 60, and in Fig 2 we present the corresponding life expectancies for randomly formed black couples. The figures are labeled by the race of the wife, but interracial marriages are rare for these cohorts. For all years, the percentage of white married women aged 60 who are married to white men exceeds 93 percent, and has exceeded 96 percent since 1950. For black married women aged 60, the percentage married to black men has exceeded 91 percent for all years since 1950; in 1930 and 1940 the percentage was slightly lower at 89 and 88 percent.

The effect of the increasing life expectancy of older men is evident in the figures. Until 1980, a white woman of age 60 who outlived her husband could expect to spend as much of her remaining life in widowhood as years with her husband. From 1930 to 1980, joint life expectancy for white couples increased from 10.0 years to 14.0 years, and her survivor life expectancy followed a similar pattern, increasing from 10.9 years to 13.4 years. After 1980, the joint life expectancy of white couples continued to increase, while her survivor life expectancy fell. By 2010, joint life expectancy exceeded her survivor life expectancy by 5 years–joint life expectancy was 17.7 years and her survivor life expectancy was 12.5 years. For married black women of age 60, joint life expectancy did not exceed her survivor life expectancy until 2010. Between 1930 and 2000, joint life expectancy for black couples increased from 8.4 years to 13.4

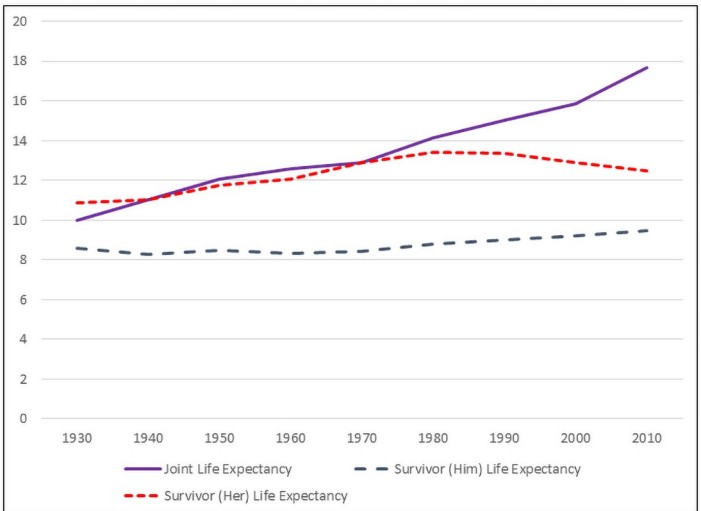

**Fig 1. Joint and survivor life expectancies, 1930–2010.** White couples, wife is aged 60.

years and her survivor life expectancy increased from 11.3 years to 13.3 years. In 2010, joint life expectancy (15.5 years) finally exceeded her survivor life expectancy (13.5 years).

The survivor life expectancy of the husbands of 60 year old women has slowly increased between 1930 and 2010. In 1930, a white (black) husband who outlived his wife could expect to live 8.5 years (8.7 years) as a widower. By 2010, the survivor life expectancy for white (black) husbands increased to 9.5 years (10.1 years).

The probability that the wife will be the surviving spouse follows an inverted u-shaped pattern similar to her survivor life expectancy. In 1930, the probability that a 60 year old white woman would outlive her husband was 0.56. This probability increased steadily, reaching a peak of 0.69 in 1980 before falling to 0.63 in 2010. The pattern for black women is similar: the probability of being the surviving spouse was 0.58 in 1930, rose to 0.70 in 1990, and then fell to 0.63 in 2010.

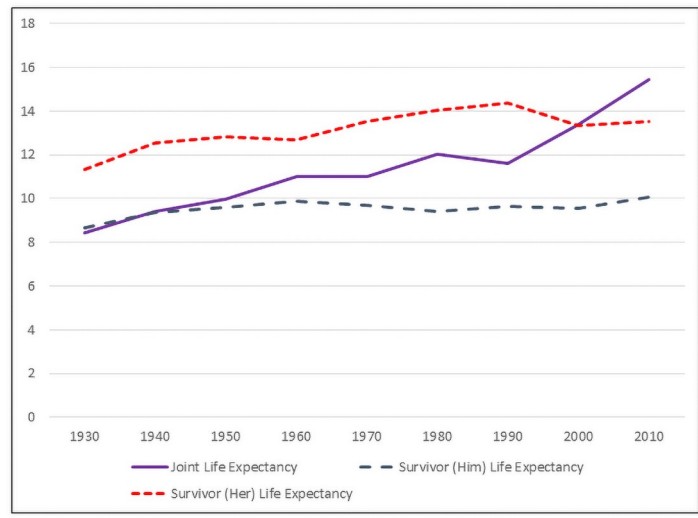

**Fig 2. Joint and survivor life expectancies, 1930–2010.** Black couples, wife is aged 60.

Two factors underlie these observed patterns. In recent decades, the life expectancy of older men has increased faster than the life expectancy of older women, implying an increase in joint life expectancy and a decrease in her survivor life expectancy. The age gap between spouses has also fallen, and fell markedly between 2000 and 2010, but the change in the life expectancies of older men and women, not changes in the age gap, are the primary drivers of the observed patterns. We performed counterfactual calculations of joint life expectancy holding the age gap fixed at its 2010 level. Because the average age gap exceeds two years in all census years except 2010, replacing the actual age gaps in each year with the 2010 age gaps raises joint life expectancy and lowers survivor life expectancy. With one exception, however, the decade-to-decade patterns noted above remain.

## Dispersion

Of course, like individual life expectancies, joint and survivor life expectancies are summary measures of age-specific mortality distributions and provide no information about their dispersion. To provide information about the dispersion of the mortality distributions around the joint life expectancy and survivor life expectancies, we return to our focal couple (a 60-year-old non-Hispanic white woman and a 62-year-old non-Hispanic white man) and calculate the probability that this randomly formed couple will survive ten, twenty, thirty, or forty years. We present the results in a 4×4 matrix (Table 3). The matrix shows the probabilities of various realizations of the timing of death for the couple and the surviving spouse. The four corner cells, which are the four lowest probabilities in the matrix, correspond to the cases in which one or both spouses die in either the first decade or the last decade. For example, the top right cell, (1, 4), corresponds to the case in which the husband dies within the first decade and the wife lives at least until age 90; thus, the probability that the wife will be a widow for more than 30 years is 0.034. In cells above the principal diagonal, the wife is the surviving spouse; in cells below, the husband is the surviving spouse; in the four cells on the principal diagonal, surviving spouse may be the wife or the husband. If mortality were evenly distributed across the 16 cells of the matrix, each would have a probability of 0.0625. The most likely scenario is that both spouses live between 20 and 30 additional years, but this probability is only 0.129.

The wide dispersion of these distributions owes little to the differing mortality rates of men and women or to the age gap between spouses, but reflects the large overlap of the underlying mortality distributions. If we perform the same calculation for a same-sex couple consisting of two 60-year-old women or two 62-year-old men, the resulting matrix is very similar. The wide dispersion in mortality rates around joint and survivor life expectancies suggests the need for caution when interpreting summary measures such as joint life expectancy and survivor life expectancy, but similar caution is also required when interpreting individual life expectancy.

**Table 3. Distribution of expected mortality.** Wife is aged 60, he is aged 62.

| HUSBAND | | WIFE | | | | |
|---|---|---|---|---|---|---|
| | | 61–70 | 71–80 | 81–90 | 91–100 | Total |
| | 63–72 | 0.031 | 0.057 | 0.078 | 0.034 | 0.200 |
| | 73–82 | 0.045 | 0.084 | 0.114 | 0.050 | 0.293 |
| | 83–92 | 0.051 | 0.094 | 0.129 | 0.056 | 0.331 |
| | 93–102 | 0.027 | 0.050 | 0.068 | 0.030 | 0.175 |
| | Total | 0.154 | 0.285 | 0.390 | 0.170 | 1.000 |

## Conclusion

We have defined and calculated measures of the joint and survivor life expectancies of couples. Although it is intuitively appealing to calculate couple-based measures of life expectancy directly from men's and women's life expectancies, such calculations overstate joint life expectancy and dramatically understate survivor life expectancy of older couples because they fail to take account of the substantial overlap in spouses' mortality distributions. We calculate joint and survivor life expectancy for randomly formed older couples using 2010 data disaggregated by race and ethnicity and by education, and for white and black couples from 1930–2010. Although the measures are tedious to calculate, tools such as the "Life Expectancy Calculator" on the Social Security website [31] could easily be augmented to calculate couples' joint and survivor life expectancies.

Joint and survivor life expectancies provide a more complete picture of the life course path faced by married couples. Comparisons by race and ethnicity, and by education, show how even without accounting for correlations, patterns of assortative marriage on race and education amplify differences in life expectancies. Viewing the measures over time, we observe how increases in the life expectancy of one gender relative to the other affects couples' joint life expectancy and the life expectancy of the surviving spouse. Social science research that focuses on retirement, long-term care and other issues involving older individuals may benefit from incorporating joint life expectancies into the discussion. For example, recent discussions of premature mortality of white men ("deaths of despair") may consider how a decline in the life expectancy of men impacts the expected length of widowhood, and how this in turn may impact policy.

Our aim is not to provide an in-depth analysis of the determinants of joint and survivor life expectancies across race and ethnicity, education or time. Because we base our calculations on individual life tables, we cannot take account of the possibilities that (a) healthier individuals may select into marriage, (b) marriage may itself increase life expectancy, and (c) mortality rates of husbands and wives may be correlated. Because taking account of these possibilities requires richer data, we leave these refinements for future research. Our goal is to simply highlight the measures and to note that these calculations–even with their limitations–are as important as benchmarks for retirement and long-term healthcare decisions as are individual life expectancies.

## Acknowledgments

We are grateful to Magali Barbieri, Itzik Faldon, Claudia Goldin, Larry Katz, Andrew Noymer, James Poterba, Peter Wiedenbeck, Robert Willis, and Jeffrey Zax for their comments and suggestions. We are also grateful to participants in the "Women Working Longer" project and those in the MRRC's Researcher Workshop, the PAA in Austin, the NBER Cohort Studies workshop, and the Michigan Health and Retirement symposium for their comments and suggestions.

## Author Contributions

**Writing – original draft:** Janice Compton, Robert A. Pollak.

**Writing – review & editing:** Janice Compton, Robert A. Pollak.

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
