## [Decision Letter · Decision Letter 0]

24 Sep 2020

PONE-D-20-17148

The Life Expectancy of Older Couples And Surviving Spouses

PLOS ONE

Dear Dr. Pollak,

Thank you for submitting your manuscript to PLOS ONE. After careful consideration, we feel that it has merit but does not fully meet PLOS ONE’s publication criteria as it currently stands. Therefore, we invite you to submit a revised version of the manuscript that addresses the points raised during the review process.

The reviewers and I see great merit in your work. However, there are several areas where the manuscript can be improved prior to publication.  As you will see, the reviewers were in agreement about the importance of your contribution, and offer a number of different ways to strengthen your work (e.g., reorganizing your paper, rounding out the literature review, explicating the methodological approach in more detail, etc.). 

My read of your work is that you offer a compelling approach for how couple decisions around retirement and widowhood matter in life-course research.  In addition to the reviewer comments, there are two additional aspects that I believe would strengthen your manuscript. 

First, it would be helpful to step back and re-frame your paper.  Although your empirical inquiry is of importance to the Social Security Administration, one could easily think of other reasons that might motivate this inquiry.  For instance, it could be that "deaths of despair" have totally shifted ages of death for white men of lower SES in ways that will affect the individual, joint, and survivor life-expectancies.  I would suggest that you draw out the social significance of your inquiry by, potentially, framing your inquiry around Case & Deaton's work on increases in premature mortality among white men. 

Similarly, the period of your study (2010) is right on the tail-end of the Great Recession (2009), which certainly would affect retirement decisions and wealth accumulation.  You may want to engage or theorize how the Great Recession affected retirement decisions among men and women in the age group that you study. 

Second, your findings on race and education are surprising, in large part, because conventional wisdom is that black men have a lower life-expectancy than white men.  The same is true for Black women.  Yet, to find that marriage somehow leads to higher survivor life-expectancy among blacks than whites suggests that the classic economic model of gains to marriage for black men and women need to be reconceptualized to account for mortality differentials.  You may want to discuss these reversals in the context of marital gains. 

Lastly, you may want to rewrite your abstract.  I don't think lines 48 (beginning with "To illustrate,...") through 56 are necessary.  Instead, I would like to encourage you to discuss your method and general findings (including race & education) in the abstract. 

We look forward to receiving your revised manuscript.

Kind regards,

Bryan L. Sykes, Ph.D.

Academic Editor

PLOS ONE

Journal Requirements:

Reviewers' comments:

Reviewer's Responses to Questions

**Comments to the Author**

1. Is the manuscript technically sound, and do the data support the conclusions?

Reviewer #1: Yes

Reviewer #2: Yes

2. Has the statistical analysis been performed appropriately and rigorously? 

Reviewer #1: Yes

Reviewer #2: Yes

3. Have the authors made all data underlying the findings in their manuscript fully available?

Reviewer #1: No

Reviewer #2: Yes

4. Is the manuscript presented in an intelligible fashion and written in standard English?

Reviewer #1: Yes

Reviewer #2: Yes

5. Review Comments to the Author

Reviewer #1: I like the paper and find the topic important. As stated in my report, I do not always find it easy to follow the estimations in the paper. My report suggests a restructuring and gives examples of where I find the paper unclear.

Reviewer #2: This is a very interesting and well written paper that I enjoyed reading. I think however that the authors need to address some additional comments before publication.

1. All analyses are done using period life expectancy data and referring to period life expectancy. It is important to discuss the relationship between period and cohort life expectancy and to specifically address how their period life expectancy estimates relate to cohort life expediencies (which for the cohorts under consideration is still unknown, but can be estimated).

2. I would like to see a discussion of the interdependence of wife’s and husband’s mortality. For instance, a large body of research shows that widowhood results in an increased risk of death, specifically for men and that is particularly high in the first year after the spouse dies. Although the authors correctly point out that “the couple life cycle models must be substantially complex”, this interdependence is not mentioned. This interdependence is not addressed and I’d like to see a discussion how the authors’ estimates, and specifically the survivor life expectancy, are affected by this interdependence in life expectancy.

3. The observed period life expectancy (used in the analyses) is a mixture of the mortality of married and unmarried individuals. As a result, for married individuals the period life table overestimates mortality, while for unmarried individuals it underestimates. This has implications for the calculated joint life expectancy (underestimating it) and can substantially shift the results. The authors need to address this important issue and discuss how it affects their results.

Points 3 and 4 are very briefly mentioned in the text (lines 193-201), but how these isues affect the authors’ estimates is not discussed. I recommend to expand this discussion and address implications for your results.

4. I am also wondering why the analyses are based on 2010 data and are not using a more recent dataset reflecting more recent trends in the US mortality.

Some additional comments:

Clarify in the abstract which data source is used for this analysis.

The abstract should also mention that the paper investigates differences in joint and survivor life expectancy by race and ethnicity and education.

Clarify how does your approach differ from what Goldman and Lord have done in the past. Related to this, clarify what is exactly the contribution of this paper and how do the present results change our understanding/thinking, what are the implications of these results. The authors point that the concept of joint life expectancy has not bee widely used, and instead age specific mortality rates are widely used in different applications. Why should we instead start using joined life expectancy? Why is the concept presented in this paper advantageous? A broader discussion of these topics will better motivate the paper.

Also, I suggest to move the text in lines 181-201 into the introduction; it is somewhat misplaced in the methods section.

Line 229 – why is race/ethnicity of couples defined by race of wife? If race of husband is used instead, will this change the results?

The differences in joint and survivor life expectancy by race and education are striking, but the implications of these differences are not discussed at all. Expanding this discussion will contribute to the relevance of the paper.

6. PLOS authors have the option to publish the peer review history of their article (what does this mean?). If published, this will include your full peer review and any attached files.

Reviewer #1: No

Reviewer #2: No

---

## [Author Response · Author response to Decision Letter 0]

16 Feb 2021

Editor’s Comments

First, it would be helpful to step back and re-frame your paper. Although your empirical inquiry is of importance to the Social Security Administration, one could easily think of other reasons that might motivate this inquiry. For instance, it could be that "deaths of despair" have totally shifted ages of death for white men of lower SES in ways that will affect the individual, joint, and survivor life-expectancies. I would suggest that you draw out the social significance of your inquiry by, potentially, framing your inquiry around Case & Deaton's work on increases in premature mortality among white men. 

In the introduction, we included a point on how the consideration of joint and survivor life expectancies could add to the discussion of inequalities by race and ethnicity, and by education. In particular, see lines 112-121. We also note in the conclusion (lines 427-430) that even without accounting for correlations, patterns of assortative mating by race and education amplify differences in life expectancies. 

We agree that there is an important dimension of the “deaths of despair” that is missed when the focus is on the individual mortality, namely the impact that individual mortality has on the life course of others in the household. We note this in the conclusion (lines 430-436) as an avenue for future research. However, we did not discuss the issue in detail for two reasons. First, we focus on women who are married at age 60 and their husbands. Given the age and marital status of our focus, we are missing the vast majority of those affected by the increased mortality due to drug overdose and suicide. Second, our focus in this paper is to highlight the importance of joint and survivor life expectancies and the broad trends. It is beyond the scope of this paper to analyze or speculate on the social and economic determinants of the trends and we leave that for future research. 

Similarly, the period of your study (2010) is right on the tail-end of the Great Recession (2009), which certainly would affect retirement decisions and wealth accumulation. You may want to engage or theorize how the Great Recession affected retirement decisions among men and women in the age group that you study. 

You are certainly correct that the Great Recession would impact the retirement decisions of the couples in whom we are interested. Indeed, this paper is an offshoot from a project that aimed to look at the determinants of women’s labour force participation following the Great Recession. Our focus was initially on how changes in joint and survivor life expectancy alters the time horizon that older couples face when making their retirement and savings decisions. In particular, we noted that there are two horizons to consider when making these decisions - the length of time expected to be alive together, and the length of time expected to be a widow or widower. Our goal here is therefore to explain and document the benchmarks of what these time horizons would look like for randomly formed couples – and to note that the back of the envelope calculations that one might rely on, using individual life expectancies, are quite wrong. 

Second, your findings on race and education are surprising, in large part, because conventional wisdom is that black men have a lower life-expectancy than white men. The same is true for Black women. Yet, to find that marriage somehow leads to higher survivor life-expectancy among blacks than whites suggests that the classic economic model of gains to marriage for black men and women need to be reconceptualized to account for mortality differentials. You may want to discuss these reversals in the context of marital gains. 

We are glad you found the race and education numbers intriguing. However, the higher survivor life expectancy among black women is not related to the gains to marriage. Our calculations are simply capturing the effect of overlapping age-specific mortality rates of men and women. The survivor life-expectancy of black women is due to the higher gap between mortality of black women and men, compared to gap in mortality of white men and women. In other words, black women can expect longer widowhood. 

Lastly, you may want to rewrite your abstract. I don't think lines 48 (beginning with "To illustrate,...") through 56 are necessary. Instead, I would like to encourage you to discuss your method and general findings (including race & education) in the abstract. 

Thank you. We have substantially revised the abstract with your suggestions. 

Reviewers' comments:

Have the authors made all data underlying the findings in their manuscript fully available?

Reviewer #1: No

Reviewer #2: Yes

The data that we use in the paper is publicly available and noted in references number [16-17]. 

Reviewer 1:

All analyses are done using period life expectancy data and referring to period life expectancy. It is important to discuss the relationship between period and cohort life expectancy and to specifically address how their period life expectancy estimates relate to cohort life expediencies (which for the cohorts under consideration is still unknown, but can be estimated).

Our goal for these calculations was to provide benchmark figures for joint and survivor life expectancies. We kept the calculations as close to the individual life expectancy calculations that are most used, using age-specific mortality rates. We note in lines 130-135 the issue of period and cohort life expectancy calculations however we leave the specific calculations to further research. 

I would like to see a discussion of the interdependence of wife’s and husband’s mortality. For instance, a large body of research shows that widowhood results in an increased risk of death, specifically for men and that is particularly high in the first year after the spouse dies. Although the authors correctly point out that “the couple life cycle models must be substantially complex”, this interdependence is not mentioned. This interdependence is not addressed and I’d like to see a discussion how the authors’ estimates, and specifically the survivor life expectancy, are affected by this interdependence in life expectancy.

We deliberately exclude interdependencies with the construction of random couples. Our calculations are by design average measures based on random matches of individuals identified only by age, sex, race and education. We do this so that these benchmark figures are analogous to the standard individual life expectancies. Just as the standard life expectancy does not account for an individual’s choices made over their lifetime (which may lead one to a higher or lower life duration than the average) the population based joint and survivor life expectancies do not account for the choice of which particular spouse to marry, nor to the life choices of the spouses once they are married. We discuss this in the text lines 149-156. 

The observed period life expectancy (used in the analyses) is a mixture of the mortality of married and unmarried individuals. As a result, for married individuals the period life table overestimates mortality, while for unmarried individuals it underestimates. This has implications for the calculated joint life expectancy (underestimating it) and can substantially shift the results. The authors need to address this important issue and discuss how it affects their results.

This is true. However, it would not be appropriate to use the mortality rates of married individuals rather than the mortality rates of the full population to calculate life expectancies of married individuals. To do so would assume that the individual is married until death, which is rarely true of both spouses. To account for the differences by married/not, we would need longitudinal data on married couples at age 60 and followed forward until the deaths of both. We leave this for future research, as our goal for this paper was to provide benchmark estimates using the data that are easily available. Moreover, it should be noted that the annual mortality rates of married and unmarried individuals may be capturing the mortality effects of other characteristics that are correlated with both marriage and mortality, and not necessarily reflecting an effect of marriage per se. We discuss this in the text in lines 136-148.

Points 3 and 4 are very briefly mentioned in the text (lines 193-201), but how these isues affect the authors’ estimates is not discussed. I recommend to expand this discussion and address implications for your results.

We expand on these points in the introduction, but are unable to say how exactly these issues affect the estimates. Our estimates in this paper are meant to provide a baseline which can then be compared with refined estimates based on couple rather than individual data. 

I am also wondering why the analyses are based on 2010 data and are not using a more recent dataset reflecting more recent trends in the US mortality.

We have added in a section that compares the data from decennial years over time (lines 320-385). This was the initial plan of the paper, and this is why we focussed on 2010 for the cross-section comparisons – it is the latest decennial year available. Rather than discuss deviations due to economic conditions, etc., our goal is to first document long-term trends in joint and survivor life expectancies. 

Clarify in the abstract which data source is used for this analysis. The abstract should also mention that the paper investigates differences in joint and survivor life expectancy by race and ethnicity and education.

Thank you. We have substantially altered the abstract to address the comments of the referees and the editor. 

Reviewer 2:

Clarify how does your approach differ from what Goldman and Lord have done in the past. Related to this, clarify what is exactly the contribution of this paper and how do the present results change our understanding/thinking, what are the implications of these results. The authors point that the concept of joint life expectancy has not bee widely used, and instead age specific mortality rates are widely used in different applications. Why should we instead start using joined life expectancy? Why is the concept presented in this paper advantageous? A broader discussion of these topics will better motivate the paper.

The approach is not different, our contribution is to highlight how the approach can be applied so that differences over time, by race and ethnicity, and by education provide a fuller picture of joint and survivor LE. We should not consider “Male LE” or “Female LE” in isolation because many decisions are made jointly. Also, when we consider by race and ethnicity, and education, we can see that inequality in individual LE gets compounded into inequalities in joint and survivor LE. 

Also, I suggest to move the text in lines 181-201 into the introduction; it is somewhat misplaced in the methods section.

Thank you for this suggestion. We have substantially altered the introduction to clarify the calculations in the paper, and have moved this text as per your suggestion. 

Line 229 – why is race/ethnicity of couples defined by race of wife? If race of husband is used instead, will this change the results?

The proportion of older couples who are mixed race is small, whether they are anchored on the husband or the wife. In 2010, 96 percent of married 60 year old non-Hispanic white women have non-Hispanic white husbands; while 94 percent of married 62 year old non-Hispanic white men have non-Hispanic white husbands. The corresponding figures for black and Hispanic couples are lower: 91 percent of black women are married to black men and 88 percent of black men are married to black women; 76 percent of Hispanic women are married to Hispanic men and 78 percent of Hispanic men are married to Hispanic women. 

We provide information for only same race/ethnicity couples to highlight assortative marriage but, as with all the summary numbers provided, we are admittedly not capturing all factors that affect life expectancies. The proportion of couples that are represented in the summary measures does not depend on whether we anchor race/ethnicity on the husband or the wife. 

The differences in joint and survivor life expectancy by race and education are striking, but the implications of these differences are not discussed at all. Expanding this discussion will contribute to the relevance of the paper.

See above. We add brief discussions of this issue in the introduction and conclusion. A full analysis of the implications of this is beyond the scope of this paper. 

Reviewer 3:

Generally, the empirical analysis seems convincing, although it is not always quite clear what theauthors have done in every step, and how the data was used.

Thank you for this comment. We have substantially altered the description of the method. 

I think the paper would be more interesting to read if it more clearly explained early on in the paper, perhaps in the Introduction, why we would expect that it is important to take within couple mortality correlations into account. The background and previous literature on intramarriage correlations and sorting into marriage is not introduced and discussed before 7 pages into the paper (lines 181-195), and the literature survey is very short. The paper would improve by extending the literature discussion on why we should expect intra-spouse correlation in health and mortality, citing papers on 1) assortative mating in health, wealth etc. (e.g. Guner,Kulikova & Llull in European Economic Review, Vol. 104, May 2018) and selection into marriage, intra-marriage correlation in health behavior through habits, smoking, drinking, exercise, diet etc., as well as information exchange within couples (e.g. Fadlon and Nielsen, 2019, American Economic Review, “Family Health Behaviors”), 3) coordination of couples’ retirement decisions that may impact health and life expectancy (e.g. Gustman and Steinmeier 2000 in Journal of Labor Economics). This section may also include a short discussion of health effects of bereavement of a spouse.

In this new version of the paper we are more clear in noting that we are specifically not taking into account the within couple mortality correlations. While we believe it is important to address these correlations in future research but we first need a clean slate benchmark measure of couple life expectancies which do not incorporate these correlations. The consideration of how couples’ realized death patterns are correlated requires longitudinal data and is necessarily backward looking. Our aim here is to construct measures that are similar to the individual life expectancy measures and so are based on population averages and do not incorporate individual couple deviations.

The structure of the Methods section is a bit confusing. It starts out with a very brief mention of the data, then goes on to present the idea of constructing life expectancies 0based on individual mortality data and showing some results based on this approach in Table 1, then moves to a short literature discussion (mentioned in point 1 above), and then finally presents the methods used. A restructuring of this section would be helpful.

 We have restructured this and hope it is more clear. 

Following point 2), I suggest that the authors create a specific Data section explaining in a bit more detail about the 2010 Census and NCHS data for readers who are not quite familiar with these datasets. It was not clear to me whether the authors had access to information at the individual for both partners in a couple? Also, what is the source of the life expectancies from these data? Maybe it is stated in another part of the paper.

In this version, we have been more careful to explain that the couples for which we are calculating joint and survivor life expectancies are randomly formed couples. The life expectancies are calculated from the individual mortality rates. 

It is not clear to me what data are used to create Table 1. Moreover, in general, tables should be self-explanatory, without having to read the text in detail. I suggest making a note to Table 1 with instruction about how to read it. Moreover, in line 197: When speaking about “Our calculations, …”, I presume that the authors are speaking about the calculations leading to Table

1?

We have moved this discussion to the end, and believe that in this section, the construction of this table now makes more sense. Thank you for this point. 

Methodological contribution: In order to assess the significance of taking proper account of

intra-couple correlation in health and mortality rather than just using individual data, would it be possible e.g. to generate a table like Table 1, but just based on the data for couples?

We have tried to be more clear that we are not using couple data, but rather forming couples randomly and applying individual mortality data. 

I found the last four lines of the Conclusion confusing, especially point c), as I thought that the point of the whole exercise in the paper is to show the importance of taking marital status into account when calculating life expectancies for couples.

We hope the goal of this paper is more clear in this version. 

Specific and minor comments

Lines 94-98 in the Introduction more or less repeat the text in lines 65-68.

• Thank you. We have changed the introduction substantially based on the comments provided.

---

## [Decision Letter · Decision Letter 1]

12 Apr 2021

The Life Expectancy of Older Couples And Surviving Spouses

PONE-D-20-17148R1

Dear Dr. Pollak,

We’re pleased to inform you that your manuscript has been judged scientifically suitable for publication and will be formally accepted for publication once it meets all outstanding technical requirements.

Kind regards,

Bryan L. Sykes, Ph.D.

Academic Editor

PLOS ONE

Additional Editor Comments (optional):

Reviewers' comments:

Reviewer's Responses to Questions

**Comments to the Author**

1. If the authors have adequately addressed your comments raised in a previous round of review and you feel that this manuscript is now acceptable for publication, you may indicate that here to bypass the “Comments to the Author” section, enter your conflict of interest statement in the “Confidential to Editor” section, and submit your "Accept" recommendation.

Reviewer #1: All comments have been addressed

2. Is the manuscript technically sound, and do the data support the conclusions?

Reviewer #1: Yes

3. Has the statistical analysis been performed appropriately and rigorously? 

Reviewer #1: Yes

4. Have the authors made all data underlying the findings in their manuscript fully available?

Reviewer #1: Yes

5. Is the manuscript presented in an intelligible fashion and written in standard English?

Reviewer #1: Yes

6. Review Comments to the Author

Reviewer #1: I think the revised manuscript has improved substantially. My main concerns with the prior version were that the structure of the paper was confusing, the description of the data unclear, and the contribution not particularly well articulated. I think the authors have taken very well care of all my comments. The paper now reads well, and the main message is clearly communicated. The paper thus provides a nice contribution to the literature on life expectancy in couples.

7. PLOS authors have the option to publish the peer review history of their article (what does this mean?). If published, this will include your full peer review and any attached files.

Reviewer #1: No

---

## [Editor Report · Acceptance letter]

6 May 2021

PONE-D-20-17148R1 

The life expectancy of older couples and surviving spouses 

Dear Dr. Pollak:

I'm pleased to inform you that your manuscript has been deemed suitable for publication in PLOS ONE. Congratulations! Your manuscript is now with our production department. 

Kind regards, 

on behalf of

Dr. Bryan L. Sykes 

Academic Editor

PLOS ONE